# Phosphate Control in Peritoneal Dialysis Patients: Issues, Solutions, and Open Questions

**DOI:** 10.3390/nu15143161

**Published:** 2023-07-16

**Authors:** Valeria Cernaro, Michela Calderone, Guido Gembillo, Vincenzo Calabrese, Chiara Casuscelli, Claudia Lo Re, Elisa Longhitano, Domenico Santoro

**Affiliations:** Unit of Nephrology and Dialysis, Department of Clinical and Experimental Medicine, University of Messina, 98125 Messina, Italy; mic.cald13@gmail.com (M.C.); guido.gembillo@unime.it (G.G.); v.calabrese@outlook.it (V.C.); chiara.casuscelli88@gmail.com (C.C.); loreclaudia96@gmail.com (C.L.R.); elisa.longhitano@libero.it (E.L.); dsantoro@unime.it (D.S.)

**Keywords:** diet, hyperphosphatemia, peritoneal dialysis, phosphate, phosphate binders

## Abstract

Hyperphosphatemia is a common complication in advanced chronic kidney disease and contributes to cardiovascular morbidity and mortality. The present narrative review focuses on the management of phosphatemia in uremic patients receiving peritoneal dialysis. These patients frequently develop hyperphosphatemia since phosphate anion behaves as a middle-size molecule despite its low molecular weight. Accordingly, patient transporter characteristics and peritoneal dialysis modalities and prescriptions remarkably influence serum phosphate control. Given that phosphate peritoneal removal is often insufficient, especially in lower transporters, patients are often prescribed phosphate binders whose use in peritoneal dialysis is primarily based on clinical trials conducted in hemodialysis because very few studies have been performed solely in peritoneal dialysis populations. A crucial role in phosphate control among peritoneal dialysis patients is played by diet, which must help in reducing phosphorous intake while preventing malnutrition. Moreover, residual renal function, which is preserved in most peritoneal dialysis patients, significantly contributes to maintaining phosphate balance. The inadequate serum phosphate control observed in many patients on peritoneal dialysis highlights the need for large and well-designed clinical trials including exclusively peritoneal dialysis patients to evaluate the effects of a multiple therapeutic approach on serum phosphate control and on hard clinical outcomes in this high-risk population.

## 1. Introduction

According to the Kidney Disease: Improving Global Outcomes (KDIGO) guidelines [1], chronic kidney disease (CKD) is defined by a decrease in renal function (glomerular filtration rate < 60 mL/min/1.73 m^2^) and/or the presence of one or more markers of kidney damage that may include albuminuria, urine sediment abnormalities, electrolyte, or other abnormalities caused by tubular disorders, histological alterations at the renal biopsy, structural abnormalities showed by imaging, or history of renal transplantation.

CKD is a global health burden, with an estimated worldwide prevalence between 11 and 13% due to population aging as well as an increase in arterial hypertension and diabetes mellitus prevalence [2]. Decreased renal function is regarded as an independent risk factor for death, cardiovascular disease, and hospitalization [3] and is associated with poor quality of life and cognitive decline [4].

Diet is a mainstay in the management of any stage of chronic kidney disease (CKD). In patients under conservative therapy, the purposes include preventing and treating the metabolic consequences of progressive kidney dysfunction and delaying renal death; in subjects requiring renal replacement therapy, a proper dietary regimen contributes to ameliorating the acid–base balance, electrolytic abnormalities, and fluid overload in association with dialysis, while maintaining an adequate nutritional status.

Phosphate control is a remarkable clinical issue due to the involvement of hyperphosphatemia and hyperparathyroidism in the pathophysiology of cardiovascular disease in this population. The cited alterations are part of a syndrome, defined as chronic kidney disease–mineral bone disorder (CKD-MBD) by the 2006 KDIGO guidelines [5], which can comprise one or more of the following manifestations: abnormalities in calcium, phosphate, parathyroid hormone, or vitamin D metabolism; altered bone volume, mineralization, linear growth, strength, or turnover; extraskeletal calcifications. In particular, high serum phosphate favors endothelial dysfunction [6] and vascular calcification development [7], acting as an independent CKD-specific risk factor for cardiovascular disease in nondialysis as well as in dialysis-dependent CKD patients.

The impact of high serum phosphorous on vascular disease is clear both in subjects with normal kidney function and in those affected by moderate CKD [8,9]. The role of hyperphosphatemia in the pathophysiology of vascular calcification in patients with end-stage renal disease is deemed to be different between diabetic and nondiabetic patients. In more detail, high serum phosphate is significantly associated with vascular calcification in dialysis patients without diabetes [10] whereas it seems not to be a predictor of this alteration in diabetic dialysis patients, among which poor glycemic control has a major role [11].

Furthermore, high serum phosphate and secondary hyperparathyroidism appear to be related to lower hemoglobin concentration [12]. A very recent experimental study suggests that hyperphosphatemia induces inflammation with increased hepcidin synthesis in the liver and resulting anemia [13].

According to the described role of hyperphosphatemia in the pathophysiology of vascular calcification and other complications of CKD, many efforts are made to control phosphate and mineral metabolism alterations in patients with impaired renal function.

Diet, dialysis, and phosphate-lowering pharmacological agents are the three pillars on which the KDIGO 2017 Clinical Practice Guideline Update for CKD-MBD establishes the management of hyperphosphatemia [14].

The present narrative review focuses on the management of high serum phosphate in patients with end-stage renal disease receiving peritoneal dialysis.

## 2. Effects of Phosphate Burden and Pathophysiology of CKD-MBD

Serum phosphate levels tend to be higher than the upper limit of the normal range only in advanced CKD. However, the abnormalities leading to CKD-MBD development start from the early stages of renal function impairment, primarily in relation to an increased phosphate burden. It is deemed that the first alteration is represented by a decrease in the expression of klotho, a coreceptor playing a crucial role in the peripheral actions of fibroblast growth factor 23 (FGF23). Klotho is a transmembrane protein expressed in the kidney as well as in the brain, pancreas, and parathyroid glands and regulates several functions including fibrosis, apoptosis, cell senescence, and mineral metabolism. There are two different forms of klotho: one is a membrane-bound and the other is a soluble form, which is the result of the activity of specific enzymes, known as proteases, that split the extracellular domain in Kl1 and Kl2 [15].

FGF23 is a phosphaturic hormone released by osteocytes and osteoblasts in response to oral phosphate intake and increased serum levels of parathyroid hormone and 1,25-dihydroxyvitamin D3. In CKD patients, the progressive reduction in renal function determines a phosphate retention that stimulates the synthesis of FGF23, which acts in the kidneys, reducing phosphate reabsorption and increasing phosphate excretion as well as calcium and sodium reabsorption [16].

Klotho expression is stimulated by 1,25-dihydroxyvitamin D3 [17]. Accordingly, low serum concentrations of vitamin D in patients with CKD reduce klotho synthesis in the parathyroid glands and in the kidneys, thereby inducing a progressive peripheral resistance to FGF23 with a resulting increase in FGF23 production [18]. The increase in FGF23 levels at this stage represents a compensatory mechanism to high phosphate burden. Nevertheless, it takes part in the pathophysiology of CKD-MBD since it is able to inhibit 1α-hydroxylase activity in the kidneys, with a resulting decreased production of 1,25-dihydroxyvitamin D3. This induces secondary hyperparathyroidism, alterations in calcium and phosphorous serum levels, further FGF23 production, enhanced bone remodeling, and vascular smooth muscle cell calcifications [16].

The multiple factors involved in the pathophysiology of CKD-MBD play specific roles in the pathogenesis of CKD-related vascular calcification and cardiovascular alterations, contributing significantly to the high risk of cardiovascular morbidity and mortality in patients with impaired renal function.

The development of vascular calcification is a complex process involving several pathophysiological mechanisms including osteochondrogenic differentiation, vascular smooth muscle cell apoptosis, mineral deposition, and elastin degradation. According to the site of mineral deposition, vascular calcifications are classified in intimal, medial, and valvular calcifications [19]. All of them are very frequently observed in CKD patients because the activity of factors promoting vascular calcification (e.g., oxidative stress, inflammatory cytokines, uremic toxins, hyperparathyroidism, and increased serum levels of calcium and phosphate) overcomes that of active inhibitors (e.g., adenosine, osteopontin, osteoprotegerin, fetuin-A, and others). In particular, hyperphosphatemia is a strong inducer of vascular calcification in CKD and a major risk factor for cardiovascular disease and mortality, especially in patients with end-stage renal disease receiving dialysis. Phosphate takes part in the structure of hydroxyapatite and this is responsible for crystal precipitation in the vessels when the calcium/phosphate product increases. Most importantly, it participates in the signaling cascade, triggering vascular calcification development and promoting its progression [19,20].

The ability of phosphate to favor vascular calcification relies on various mechanisms involving specific sodium-dependent phosphate cotransporters, the type III sodium-dependent phosphate transporters Pit-1 and Pit-2, which are located in vascular smooth muscle cells and are implicated in their osteogenic differentiation and in the regulation of matrix mineralization induced by high serum phosphate, as demonstrated in experimental models [21,22]. Other not well-characterized receptors mediate the phosphate role in many processes such as vascular smooth muscle cell apoptosis, extracellular vesicle release, and matrix stability. Moreover, phosphate favors vascular calcification by inhibiting the differentiation of monocyte/macrophage into osteoclast-like cells, increasing FGF23 expression, and reducing klotho synthesis [23].

Klotho seems to have a protective role against vascular calcification. The likely mechanisms are the following: promotion of phosphate excretion by the kidneys due to the interaction with FGF23, slowing down of CKD progression, reduced expression of the Na/Pi cotransporter and the osteogenic transcription factor CBFA1/RUNX2 in vascular smooth muscle cells with consequent decreased phosphate intake and osteochondrogenic differentiation [23,24].

Clinical evidence also shows a strong association between vitamin D deficiency and cardiovascular risk. In particular, vitamin D deficiency leads to the dysregulation of the renin–angiotensin–aldosterone system, increases the proliferation of vascular cells, worsens vasodilation and insulin resistance, promotes procoagulant activity and hypertrophy of myocardial cells, and favors inflammation [25].

## 3. Phosphate Removal by Peritoneal Dialysis

Peritoneal dialysis represents a valid option for the treatment of uremic patients and appears to be as effective as extracorporeal dialysis in terms of clinical outcomes [26]. Approximately only 11% of patients requiring renal replacement therapy undergo peritoneal dialysis, mainly because this modality is less available compared to hemodialysis in many countries [27]. Among the most important advantages of peritoneal dialysis are the hemodynamic stability, the preservation of some residual renal function that helps in controlling metabolic alterations and maintaining fluid balance [28,29], and the fact of being more physiological than hemodialysis given that it is usually performed daily and lasts longer hours.

Solute and fluid transport through the peritoneal membrane occurs according to the three-pore model of peritoneal transport [30,31]. The small pores are interendothelial spaces of the peritoneal capillaries with a constant radius between 4.0 and 5.0 nm; they are characterized by peculiar junctional complexes and allow the transport of low-molecular weight solutes such as urea, creatinine, and glucose, accounting for 99.7% of the total pore surface area available to small molecule diffusion and for 90% of the peritoneal ultrafiltration coefficient. Large pores also seem to be represented by interendothelial clefts that are wider (radius averaging more than 15.0 nm) but much less numerous (less than 0.01% of the total amount of pores) than small pores and are responsible for the movement of macromolecules such as serum proteins across the peritoneal membrane; they also account for 8% of the peritoneal ultrafiltration coefficient. Lastly, ultrasmall pores (radius between 0.3 and 0.5 nm), corresponding to aquaporins-1 present both in the mesothelial cells and in the capillaries of the peritoneal membrane [32], act as water-only channels that contribute 1.5–2% to the peritoneal ultrafiltration coefficient driven by the presence of osmotically active molecules in the dialysis fluid (e.g., glucose or amino acids). The identification of ultrasmall pores explained why the osmotic agents do not rapidly dissipate from the peritoneal cavity to the blood through the small pores, as would be expected [31].

The mechanisms underlying solute exchange between peritoneal capillaries and dialysate include diffusion and convection, similar to what happens for hemodialysis techniques. Diffusion consists in the movement of molecules according to their concentration gradient. Convection is defined as the transport of solutes dragged by the fluid movement across the membrane [31,33].

Beyond these physical processes, peritoneal solute transport is also dependent on the peculiarities of the peritoneal membrane due to host-related factors. Indeed, contrary to the well-defined characteristics of the filters used in hemodialysis, peritoneal membrane properties as a dialysis membrane vary both among different individuals and in the same subject over time [34]. This has an impact on peritoneal solute and fluid transport with the need to personalize peritoneal dialysis prescription as regards modality, load volumes, number of exchanges during the day, dwell duration, and composition of dialysate solutions to optimize fluid control and blood depuration as well as improve clinical outcomes.

Peritoneal phosphate removal primarily occurs via diffusion due to endothelial transmembrane phosphate transporters, associated with a limited convective transport [35]. Phosphate molecular weight is only 96 Daltons [36] but its peritoneal transport is more complex than that of urea (60 Daltons) or creatinine (113 Daltons) since it is influenced by many factors, most of which are related to the properties of the phosphate anion with respect to the characteristics of the peritoneal dialysis membrane. Indeed, phosphate is negatively charged similar to the capillary walls and interstitial matrix; due to its hydrophilicity, it is surrounded by water molecules with a subsequent increase in the effective molecular weight; up to 40% of phosphate circulates in the blood as part of magnesium, calcium, and sodium salts and 15–20% is bound to proteins; phosphate does not diffuse freely across cellular membranes but requires transmembranous active transporters; lastly, it is mainly present within cells and shows a slow intra- and extracellular transfer rate [30,37,38]. All these features account for the observation of a phosphate peritoneal clearance lower than expected simply due to considering its molecular weight compared to that of small water-soluble molecules [37]. Accordingly, it is frequently difficult to achieve serum phosphate targets in peritoneal dialysis patients, especially when residual renal function declines [39], with a consequent increased risk of all-cause mortality [40].

Patient transporter characteristics and peritoneal dialysis modalities and prescriptions noticeably influence phosphate removal and, therefore, serum phosphate control.

In a prospective observational study including 380 adult peritoneal dialysis patients, of which 87 (22.9%) had hyperphosphatemia, Courivaud C. et al. [35] measured the total phosphate and peritoneal phosphate clearances as well as creatinine clearance and peritoneal creatinine transport by performing a peritoneal equilibration test (PET) by using a standard 2-litre-volume 2.27% glucose solution with a 4 h dwell. They found that the mere quantification of the dialysate to plasma (D/P) ratio for creatinine and peritoneal creatinine clearance is not sufficient to correctly evaluate phosphate removal and optimize it by modifying the dialytic prescription. Conversely, the determination of D/P phosphate categories and peritoneal phosphate clearance allowed for the better assessment of the peritoneal phosphate transport and clearance. This was supported by the association between a lower 4 h D/P ratio for phosphate (namely slower peritoneal transporter status) and reduced weekly peritoneal phosphate clearance with higher serum phosphate levels. Additionally, patients receiving continuous ambulatory peritoneal dialysis (CAPD) showed a greater peritoneal phosphate clearance compared to patients treated with automated peritoneal dialysis (APD) independent of their peritoneal phosphate transporter status. The authors suggest that peritoneal phosphate removal could be improved by increasing dialysate volume and dwell duration in CAPD, whereas in patients treated with APD, phosphate clearance depends on volume, number, and duration of dialysis cycles as well as on the addition of daytime exchanges in the dialytic prescription.

Debowska M. et al. [41] conducted an observational cross-sectional study including 154 prevalent peritoneal dialysis patients and assessed phosphate removal and clearance by CAPD, continuous cyclic peritoneal dialysis (CCPD), and APD. Patients treated with CAPD showed a greater total weekly phosphate removal compared to CCPD and APD as well as lower phosphatemia compared to APD patients. Phosphate removal was also directly related to the infused volume of dialysis fluid in all modalities whereas it did not correlate with ultrafiltration since, as already mentioned, phosphate elimination primarily occurs via diffusion. Nevertheless, patients belonging to the three groups differed from each other with regard to residual diuresis, gender, age, body surface area, and serum phosphate levels. It follows that the obtained results were partially affected by a selection bias.

Interestingly, a recent multicenter prospective cohort study conducted on 737 patients analyzed the effects of changing the peritoneal dialysis modality on phosphate and potassium serum levels. Transition from CAPD to APD was followed by an increase in serum phosphate whereas the opposite occurred after switching from APD to CAPD. Conversely, the two modalities appeared to be similarly efficient in controlling potassium (molecular weight: 39 Daltons), as a demonstration of the different behavior of the two solutes and of the likely greater effectiveness of CAPD in the management of serum phosphate levels compared to APD [42].

Hence, phosphate acts as a middle-size molecule in the setting of peritoneal dialysis and then its transport through the peritoneal membrane requires longer dwell times, such as those characterizing CAPD, to achieve equilibration [32], especially among lower transporters.

A small study including six low-average and six high-average transporters evaluated the impact of five tidal APD prescriptions on the peritoneal clearance of different solutes. Compared to a 50% tidal with an initial fill volume of 2 liters, a 50% tidal prescription with 2.2 L of initial fill volume and one complete renewal of the latter at midsession was associated with a significantly higher phosphate clearance. The most likely reason is that this schedule allows for maximizing the diffusive and osmotic gradients [43].

These and other studies suggest the importance of personalizing the peritoneal dialysis prescription in patients with inadequate control of serum phosphate and classified as lower transporters, through the adoption of CAPD modality or by extending dwell times on APD in order to increase the peritoneal phosphate clearance [44]. In particular, given that peritoneal phosphate clearance is slower compared to that of creatinine and urea, it is advisable to take into account the peritoneal phosphate clearance in addition to creatinine clearance when evaluating patient transport characteristics, especially in APD patients categorized as slow transporters and with reduced residual renal function [35].

## 4. Phosphate Binders in Peritoneal Dialysis Patients: Results from Clinical Trials

Given that phosphate removal by peritoneal dialysis is frequently insufficient, phosphate-lowering drugs represent an essential therapeutic strategy to manage high serum phosphate in patients with end-stage renal disease treated with this kind of renal replacement therapy. These medications bind phosphate derived from food in the gastrointestinal tract to allow its excretion in the stool, in this way preventing its absorption and reducing its serum concentration [45,46].

Currently available phosphate binders can be classified according to their molecular structure, with particular reference to the presence or absence of calcium [47].

Calcium-based compounds (calcium carbonate, calcium acetate) very effectively lower serum phosphate; however, these drugs have raised concern due to their ability to favor hypercalcemia and cardiovascular calcification development. Calcium-free phosphate binders (lanthanum carbonate, sevelamer carbonate, sevelamer hydrochloride) are also efficacious but their use can induce some potential adverse effects including nausea, vomiting, abdominal pain, constipation, or diarrhea [48]. A post hoc analysis [49] of a multicenter randomized study that compared the effects of calcium-based versus calcium-free phosphate binders on hard outcomes in CKD patients not on dialysis [50] showed that, in patients with stage 3–4 CKD and evidence of coronary artery calcification, sevelamer carbonate combined with a phosphorus-restricted dietary regimen seemed to have a protective role as regards all-cause mortality and dialysis initiation.

Aluminum-containing binders (aluminum hydroxide, aluminum carbonate) have a strong phosphate-lowering action but when ingested chronically, aluminum, which is excreted primarily by the kidneys, can accumulate and exert toxic effects primarily on bone, brain, and hematopoietic cells with a resulting development of osteomalacia, encephalopathy, and microcytic anemia, respectively [51].

Lastly, the new calcium-free iron-based compounds (ferric citrate, sucroferric oxyhydroxide) are effective in reducing intestinal phosphate absorption while improving iron parameters without significant evidence of risk of iron overload [52]. Ferric citrate efficacy and safety were tested in a clinical trial involving 292 patients on maintenance dialysis that were randomized to receive ferric citrate or active control (calcium acetate and/or sevelamer carbonate) according to a ratio of 2:1, respectively [53]. No differences were detected between the two arms, with a comparable reduction both in phosphate and in parathyroid hormone serum levels in the two cohorts after 52 weeks of treatment. Moreover, fewer serious adverse events were recorded in the ferric citrate group compared to the active control group. Ferric citrate also seems to be useful in increasing serum ferritin and transferrin saturation in dialysis patients, resulting in a reduced need for the use of intravenous elemental iron and erythropoietin-stimulating agents [54]. Similar efficacy and safety profile were reported with the administration of sucroferric oxyhydroxide. A randomized clinical trial performed on 707 hemodialysis and peritoneal dialysis patients demonstrated that sucroferric oxyhydroxide reduced serum phosphate similarly to sevelamer with a lower pill burden [55]. The literature data on the effects of sucroferric oxyhydroxide on iron parameters are controversial [56,57].

We searched on the PubMed database, without language or time range restrictions, for randomized clinical trials investigating the effects of phosphate binders in peritoneal dialysis patients, excluding those studies also involving patients on hemodialysis. We found that very few clinical trials have been designed to specifically evaluate the efficacy of phosphate binders solely in patients treated with peritoneal dialysis (Table 1) and the majority of the data justifying their large use in peritoneal dialysis derive from studies conducted on hemodialysis patients [58,59] or mixed populations [55,60].

Katopodis et al. [61] compared sevelamer hydrochloride to aluminum hydroxide in a small randomized crossover study including 30 patients on CAPD. No statistically significant differences were detected in decreases in serum phosphate between the two drugs. Additionally, a reduction in total cholesterol and low-density lipoprotein cholesterol was observed among patients treated with sevelamer hydrochloride.

The first large randomized controlled trial conducted in peritoneal dialysis patients evaluated the tolerability and efficacy of sevelamer hydrochloride and calcium acetate and was published in 2009 by Evenepoel et al. [62]. This multicenter open-label randomized parallel-group study enrolled 143 patients on peritoneal dialysis with phosphatemia higher than 5.5 mg/dL, of which 97 received sevelamer hydrochloride and 46 received calcium acetate. The two drugs efficaciously reduced serum phosphate and parathyroid hormone levels but calcium acetate, in contrast to sevelamer hydrochloride, induced a statistically significant increase in calcemia. The authors also found that the safety profiles were similar to those reported in previous studies involving hemodialysis patients.

Many years later, the safety and efficacy of sucroferric oxyhydroxide compared to sevelamer carbonate were assessed in a subanalysis [63] of a previously performed initial phase 3 randomized trial [55] followed by an extension study [64] and including 1055 dialysis patients. Of them, 84 were treated with peritoneal dialysis (56 had received sucroferric oxyhydroxide, 28 sevelamer) and were included in the subanalysis. Both binders decreased serum phosphate to a similar extent and their use was associated with primarily gastrointestinal adverse events. Importantly, sucroferric oxyhydroxide therapy was characterized by a higher adherence rate (91.2% versus 79.3%) and a lower pill burden (3.4 ± 1.3 versus 8.1 ± 3.7 tablets per day) compared to sevelamer. Actually, the high pill burden of patients on chronic dialysis may impair adherence to therapy and negatively affect not only phosphorus levels but also quality of life [65].

All described trials [61,62,63] investigated phosphate control as the main outcome and also evaluated the potential side effects of the examined phosphate binders. The follow-up duration was too brief, especially for the first two studies [61,62], and this undoubtedly prevented the effects on hard clinical outcomes from being considered.

The latest Cochrane meta-analysis performed on the treatment of hyperphosphatemia in patients with any stage of CKD showed no clear and absolute advantages of calcium-free phosphate binders on different clinical outcomes compared to calcium-containing compounds [66]. In the absence of unequivocal benefits of one phosphate binder over another, in a context where health expenditure is already very high, caution is recommended when using noncalcium-based compounds for which the cost/benefit ratio is unfavorable and whose use should be preferred in particular situations (hypercalcemia, presence of widespread vascular calcifications) [67].

Indeed, depending also on the large use of noncalcium-based binders, the cost of the phosphate binders per user-year increased by 67% in the US from 2008 to 2013 (all other drugs prescribed on dialysis went up by 21%) [68]. Therefore, new and well-designed clinical studies are needed to produce evidence of effectiveness that justifies the use of a more expensive type of phosphate binder.

## 5. Dietary Recommendations to Manage Hyperphosphatemia in Peritoneal Dialysis Patients

Phosphorous is found in many types of foods. Natural (also referred to as organic) dietary phosphorous is mainly contained in milk, dairy products, meat, poultry, and fish; it is slowly and less effectively absorbed (40–60%, with animal-derived phosphorous being more absorbed than that from plant sources) since it requires enzymatic reactions to be released from its carbon component before passing into the circulation as inorganic phosphate. On the contrary, the inorganic form, which corresponds to the phosphorous salts added to preserved, processed, or enhanced foods to improve their shelf life as well as smell and color, is promptly and efficiently absorbed (80–100%); this results in high bioavailability, because salts rapidly dissociate in the acid gastric environment without the need for enzymatic degradation [69,70].

Organic phosphorous is mostly bound to proteins in vivo. It follows that protein-rich foods, especially animal-based ones, are abundant in organic phosphorous and then, protein and phosphorous dietary intakes are strongly and positively related to each other [71].

The daily protein requirement is noticeably different between CKD patients not on dialysis and those in maintenance hemodialysis or peritoneal dialysis. According to the 2020 KDOQI Clinical Practice Guideline for Nutrition in CKD [72], a low-protein diet providing 0.55–0.60 g dietary protein/kg body weight/day is recommended in nondiabetic adult patients with stage 3–5 CKD not on dialysis, with the aim to delay renal death; alternatively, these patients may follow a very low-protein diet (0.28–0.43 g dietary protein/kg body weight/day) with the addition of keto acid/amino acid analogues to prevent malnutrition. Diabetic patients with stage 3–5 CKD not on dialysis should comply with a dietary protein intake of 0.6–0.8 g/kg body weight/day. Conversely, in hemodialysis and peritoneal dialysis patients with or without diabetes mellitus, a dietary protein intake of 1.0–1.2 g/kg body weight/day is suggested.

Nutritional status must be strictly monitored in peritoneal dialysis patients to avoid or at least reduce the risk of protein-energy malnutrition, a condition highly prevalent in this population (18-56% among CAPD patients) [73] and associated with poor clinical outcomes [74]. Many factors can contribute to net protein catabolism in patients receiving peritoneal dialysis, including the loss of albumin and amino acids with the dialysate, decrease in food intake caused by abdominal discomfort, underdialysis, and increased protein degradation [75].

Although there are reported cases of diets containing 0.9–1 g of proteins/kg/day without signs of malnutrition, a protein intake of ≥1.2 g/kg/day is recommended by the 2005 European Best Practice Guidelines (EBPG) for peritoneal dialysis [76]. Nutritional status monitoring is essential and must be carried out at least every 6 months. For this purpose, the following parameters are used in clinical practice: albumin, prealbumin, creatinine, protein equivalent of total nitrogen appearance (nPNA), anthropometry, dietary diaries, and subjective global assessment (SGA). Dual-energy X-ray absorptiometry (DEXA) is less adopted. As far as albumin is concerned, its trend rather than the single value must be considered with a downward trend indicating malnutrition. However, in some situations, such as patients with proteinuria, albumin is not a good index for malnutrition. In these cases, evaluating the tendency of lean body mass (LBM) and estimating the protein catabolic rate (PCR) can help. It is essential to manage any comorbidities and exclude chronic inflammation, since they can promote malnutrition and often lead to improperly increasing the prescribed protein load. For patients with evidence of malnutrition, in which the presence of potentially responsible comorbidities has been ruled out and who are not rapid transporters, changing the dialysis prescription to increase appetite or diffusion by the dialysate can be beneficial. In these cases, using intraperitoneal amino acids can be recommended. The prescription of oral supplements represents another helpful intervention.

The transition to hemodialysis is recommended in the most severe malnutrition cases that do not resolve.

With regard more specifically to phosphorus intake, the aforementioned 2020 KDOQI guidelines [72] recommend modulating dietary phosphorous intake to maintain phosphatemia in the normal range in patients with CKD stage 3 to stage 5 on dialysis, also taking into account the different bioavailability of the various phosphorous sources. The lack of precise dietary phosphorous ranges derives from the observation that serum phosphate levels are influenced not only by dietary intake but also by other factors including excretion in the urine in patients with residual renal function and exchanges between blood and bone tissue.

KDIGO nutritional recommendations for patients with end-stage renal disease treated with peritoneal dialysis indicate a limit of 4000 mg for dietary phosphorous daily intake (3000 mg in subjects >70 years old). Other international societies more generally recommend dietary phosphorous restriction if serum phosphate is greater than 5.0 mg/dL [77].

In addition to reducing the consumption of phosphorus-rich foods, the improvement in cooking methods could help in controlling serum phosphate in peritoneal dialysis patients, as suggested by a single-center prospective interventional trial including 97 prevalent patients [78]. Indeed, it has been reported that the processes of boiling or stewing food are able to reduce food phosphate content without influencing protein intake; moreover, soaking meat food in cold water for one hour before cooking further reduces the phosphorous content [79,80]. It follows that these cooking modalities may potentially decrease the phosphorous load associated with a given protein intake, so favoring a better control of serum phosphate while reducing the risk of protein malnutrition.

## 6. Discussion

A better knowledge of how serum phosphate control can be achieved in patients with end-stage renal disease treated with peritoneal dialysis may significantly impact on clinical outcomes (Figure 1). Dialysis adequacy is usually evaluated by assessing the small solute clearance, but phosphate behaves as a larger molecule despite its real molecular weight [38]. This implies that the amelioration of dialysis adequacy as it is commonly intended may not correspond to an effective improvement in clinically relevant outcomes in the long-term, since persistently high serum phosphate is notoriously related to cardiovascular morbidity and mortality. Peritoneal dialysis patients frequently show high phosphatemia due to the limited phosphate removal through the peritoneal membrane, especially in subjects classified as lower transporters [39]. Most patients are, therefore, prescribed phosphate binder medications, which are very effective in reducing dietary phosphate absorption but can cause gastrointestinal side effects and may be associated with high pill burden and low adherence to therapy [81]; in this regard, a cross-sectional study by Chiu et al. including 233 prevalent chronic dialysis patients found a median daily pill burden of 19, with phosphate binders accounting for approximately one-half of this value [82].

Data from clinical trials on the use of phosphate binders in patients in peritoneal dialysis are very scant. Nevertheless, their use in this population is supported by relevant pathophysiological reasons. Renal osteodystrophy in peritoneal dialysis patients has been described in several studies. Peritoneal dialysis-dependent patients have a greater prevalence of reduced bone turnover, defined as “adynamic bone disease”, compared to subjects receiving hemodialysis [83,84]. Some important risk factors are age, diabetes mellitus, and excessive suppression of parathyroid hormone serum levels due to calcium overload or treatment of hyperparathyroidism with high-dose vitamin D supplementation [85].

Without phosphate-binding therapy, all patients undergoing peritoneal dialysis are likely to have a positive phosphorus balance, unless they are severely malnourished. Indeed, an inappropriate dietary phosphorous restriction can promote the risk of sarcopenia or protein malnutrition. Consequently, phosphate binders help in controlling serum phosphate levels while allowing for an adequate protein intake [71].

Always keeping in mind the need to avoid protein malnutrition, a prominent role in serum phosphate control should be played by an appropriate nutritional program and monitoring, with the aim to reduce the dietary phosphorous load. It has been emphasized that phosphate intake associated with dietary proteins is low if compared to the phosphorous content of food additives. Consequently, phosphate restriction should primarily regard inorganic phosphorous, which is abundantly present in preserved and processed foods and is largely absorbed, rather than the organic phosphorous associated with dietary proteins, with the potential prevention of malnutrition [86].

Given that CKD patients undergoing maintenance hemodialysis or peritoneal dialysis very often receive multiple pharmacological treatments, it is necessary to take into account that a source of hidden phosphorus can also be represented by some more or less commonly prescribed medications such as amlodipine, lisinopril, sitagliptin, paroxetine, codeine, or oxycodone [87]. Moreover, CKD patients are commonly prescribed vitamin D supplementation. Although the aforementioned beneficial effects of vitamin D should not be overlooked, its fundamental role concerns calcium–phosphorus metabolism. In particular, treatment with 1,25 dihydroxy-vitamin D, which improves calcium and parathyroid hormone levels, increases the expression of both the sodium-dependent phosphate transport protein 2A (NPT2A) in the kidneys and the NPT2B in the intestine, promoting an increase in phosphorus levels, a mechanism counterbalanced by FGF23 [88]. In this way, active vitamin D supplementation could contribute to hyperphosphatemia and the related risks of cardiovascular disease and overall mortality. To avoid this, together with vitamin D supplementation that is important for its systemic effects, a simultaneous dietary phosphorous restriction is essential to reduce intestinal absorption independently of 1,25 hydroxyvitamin D [89].

Another relevant factor in the control of serum phosphate is residual renal function, which is present in the majority of patients receiving peritoneal dialysis, is associated with decreased morbidity and mortality as well as better quality of life [29], and significantly contributes to maintaining phosphate balance [90]. Thus, efforts to preserve residual renal function might help to control hyperphosphatemia. This aim can be achieved in peritoneal dialysis patients through different strategies: the use of renin–angiotensin–aldosterone system blockers, adequate control of blood pressure in order to prevent hypertension as well as hypotensive episodes, cautious prescription of diuretics, adoption of incremental peritoneal dialysis modality with careful monitoring of blood tests and clinical conditions, and use of icodextrin solution [29].

A critical issue deserving of further investigation in the control of hyperphosphatemia in peritoneal dialysis patients is that no specific serum phosphate targets have been reported in the literature, most probably because studies often include both hemodialysis and peritoneal dialysis populations. The identification of a defined serum phosphate range could improve the clinical management and outcomes of CKD patients treated with peritoneal dialysis. A large observational analysis [91] conducted on 31,989 adult incident dialysis patients, whose data were retrieved from the Australia and New Zealand Dialysis and Transplant (ANZDATA) Registry, showed a U-shaped curve for the risk of all-cause mortality related to serum phosphate when it is below about 3.9 mg/dL (1.25 mmol/L) or greater than about 6.2 mg/dL (1.99 mmol/L); these values are different from the range considered normal in nondialyzed patients. In a subanalysis of the same study including only peritoneal dialysis patients, the risk of all-cause mortality increased and was greater than in hemodialysis patients for phosphate values below about 3.1 mg/dL (1.00 mmol/L). Conversely, another large observational study [92] including 1662 incident peritoneal dialysis patients reported an increased risk of all-cause mortality when serum phosphate was higher than 5.5 mg/dL (1.78 mmol/L), a significantly lower value compared to that identified by Tiong MK et al.

In conclusion, considering that serum phosphate above the normal range frequently results in patients receiving peritoneal dialysis, large and well-designed randomized controlled clinical trials involving exclusively peritoneal dialysis patients and with an appropriate follow-up duration are required to evaluate the effects of an optimized multiple therapeutic approach on serum phosphate control and on hard clinical outcomes in this high-risk population. The current body of evidence does not lend itself to meta-analysis due to the scarcity of specific trials. As new data emerge, summarizing them in systematic reviews with meta-analyses would be helpful to guide nephrologists toward a proper and beneficial management of hyperphosphatemia in peritoneal dialysis patients.

## Figures and Tables

**Figure 1 nutrients-15-03161-f001:**
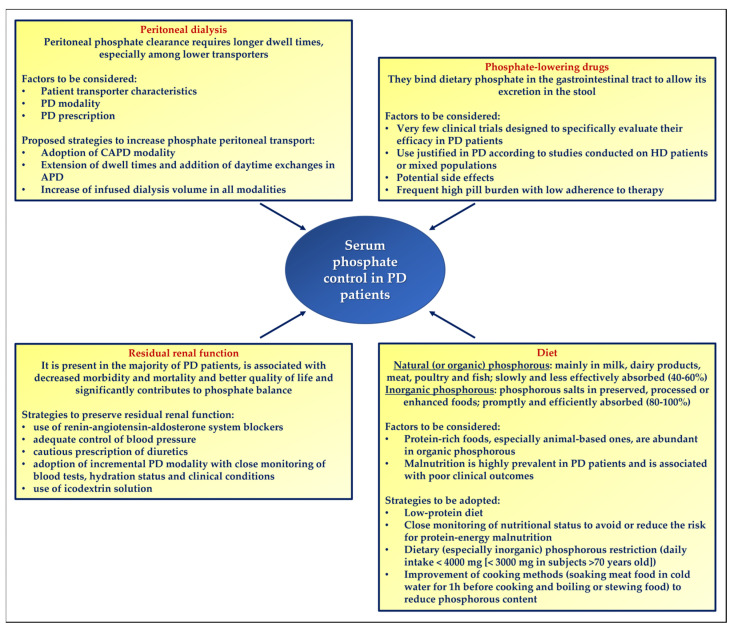
Multiple therapeutic approach for serum phosphate control in patients with end-stage renal disease receiving peritoneal dialysis. APD: automated peritoneal dialysis; CAPD: continuous ambulatory peritoneal dialysis; HD: hemodialysis; PD: peritoneal dialysis.

**Table 1 nutrients-15-03161-t001:** Clinical trials evaluating the efficacy of phosphate binders solely in patients treated with peritoneal dialysis.

Ref.	Study Design	Population	Study Drug	Control	Follow-Up Duration	Main Results
[61]	Open label, randomized crossover trial	30 adult patients on CAPD	Sevelamer hydrochloride	Aluminum hydroxide	- 2 wk phosphorus binder washout - Phase A: 8 wks - 2 wk washout - Phase B: 8 wks	- Similar reduction in serum phosphorous levels - Sevelamer hydrochloride associated with significantly reduced total cholesterol and LDL-cholesterol compared to aluminum hydroxide
[62]	Multicenter open-label randomized parallel-group trial	143 adult patients on PD (CAPD/APD 43%/57% in both groups) with serum phosphorus >5.5 mg/dL	Sevelamer hydrochloride (n. 97 patients)	Calcium acetate (n. 46 patients)	12 wks	- Serum phosphorus and PTH levels significantly reduced in both arms - Serum calcium increased in the calcium acetate group compared to sevelamer hydrochloride group - Sevelamer hydrochloride associated with decreased total cholesterol, LDL-cholesterol, and uric acid levels and increased bone-specific alkaline phosphatase levels - Hypercalcemia experienced by more patients in the calcium acetate group
[63]	Subanalysis of a previous initial phase 3 randomized trial [55] followed by an extension study [64]	84 adult patients on PD	Sucroferric oxyhydroxide (n. 56 patients)	Sevelamer carbonate (n. 28 patients)	52 wks	- Similar reduction in serum phosphate levels - Primarily gastrointestinal adverse events in both groups - Higher adherence rate to therapy and lower pill burden in the sucroferric oxyhydroxide group

Abbreviations: APD: automated peritoneal dialysis; CAPD: continuous ambulatory peritoneal dialysis; LDL: low-density lipoprotein; PD: peritoneal dialysis; PTH: parathyroid hormone; wk: week.

## Data Availability

Not applicable.

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
