# Peer review of "Phosphate Control in Peritoneal Dialysis Patients: Issues, Solutions, and Open Questions"

_nutrients, 2023, doi:10.3390/nu15143161_

Round 1
Reviewer 1 Report
The article appears to be well-written, comprehensive, and informative, providing a detailed analysis of the implications of chronic kidney disease (CKD), especially the role of diet and phosphate control in managing CKD. However, here are a few suggestions to further improve the manuscript:
1. Explanation of Complex Concepts: While the manuscript is well-structured, it includes many complex concepts that may be challenging for readers without a deep background in nephrology. Make sure to thoroughly explain these concepts and how they contribute to the larger narrative. For example, elaborate on why a decrease in Klotho expression is a significant issue, and how phosphate promotes vascular calcification.
2. Economic Implications: The cost implications of managing serum phosphate levels in peritoneal dialysis patients could provide a broader perspective, considering that costs can be a major determinant in patient and physician choices
3. Use of Visual Aids
The most critical improvement that could significantly enhance your paper's readability and comprehensibility is the inclusion of visual aids. Given the complex nature of the topic, diagrams, flowcharts, or tables could greatly aid readers in understanding the multifaceted interactions between dietary phosphate, residual renal function, dialysis, and medication in managing serum phosphate levels. For example, a detailed flow chart illustrating these interrelationships would offer a clear, visual breakdown of the process. Additionally, graphical representations of the clinical trials and their outcomes could provide an easily digestible way to convey complex data to readers who might not be experts in the field. Although other aspects of your paper might also benefit from further development, such as a more in-depth discussion on new technologies in peritoneal dialysis or the inclusion of recent studies, the priority should be the incorporation of these visual aids to ensure the clarity and effectiveness of your message.
Author Response
Point-to-point reply to Reviewer 1 - Manuscript nutrients-2460417
The article appears to be well-written, comprehensive, and informative, providing a detailed analysis of the implications of chronic kidney disease (CKD), especially the role of diet and phosphate control in managing CKD. However, here are a few suggestions to further improve the manuscript:
- Explanation of Complex Concepts: While the manuscript is well-structured, it includes many complex concepts that may be challenging for readers without a deep background in nephrology. Make sure to thoroughly explain these concepts and how they contribute to the larger narrative. For example, elaborate on why a decrease in Klotho expression is a significant issue, and how phosphate promotes vascular calcification.
We have expanded the explanation on why a decrease in Klotho expression is significant and on how phosphate promotes vascular calcification, as suggested by the Reviewer.
- Economic Implications: The cost implications of managing serum phosphate levels in peritoneal dialysis patients could provide a broader perspective, considering that costs can be a major determinant in patient and physician choices
We have added a discussion on costs of phosphate-lowering therapies at the end of the section “Phosphate Binders in peritoneal dialysis patients: results from clinical trials”.
- Use of Visual Aids. The most critical improvement that could significantly enhance your paper's readability and comprehensibility is the inclusion of visual aids. Given the complex nature of the topic, diagrams, flowcharts, or tables could greatly aid readers in understanding the multifaceted interactions between dietary phosphate, residual renal function, dialysis, and medication in managing serum phosphate levels. For example, a detailed flow chart illustrating these interrelationships would offer a clear, visual breakdown of the process. Additionally, graphical representations of the clinical trials and their outcomes could provide an easily digestible way to convey complex data to readers who might not be experts in the field. Although other aspects of your paper might also benefit from further development, such as a more in-depth discussion on new technologies in peritoneal dialysis or the inclusion of recent studies, the priority should be the incorporation of these visual aids to ensure the clarity and effectiveness of your message.
We thank the Reviewer for his/her suggestion. We have included a figure on the multiple therapeutic approach (peritoneal dialysis, residual renal function, phosphate-lowering drugs and diet) for serum phosphate control in patients receiving peritoneal dialysis. We have also added a table summarising the clinical trials evaluating phosphate binders in peritoneal dialysis patients.

Reviewer 2 Report
The primary focus of this review as outlined in the abstract is that there are special issues required regarding appropriate control of PO4 in ESKD patients on peritoneal dialysis and that the are very few studies that address directly the issue of PO4 control in PD. This reviewer endorses these questions and would support the importance of dealing with these issues.
Recommend substantial shortening of the general discussion of pathophysiology of CKD-MBD, rather focus on the impact of an increased PO4 burden in CKD (and the physiologic response to this burden) and the evidence that supports the clinical consequences of this higher PO4 burden.
A detailed discussion of the evidence used to support PO4 binder therapy in general for PD patients and any specific evidence (or lack there of) for specific target serum PO4 levels.
Detail the areas for which evidence is lacking and why such evidence developed specifically in PD patients is important.
Expand the discussion of the impact of PD prescription and of PD transport properties on PO4 clearance and the implications for treatment.
Expand the discussion of impact of diet restriction on protein malnutrition.
Include an acknowledgement that active vitamin D therapy has an impact on PO4 absorption and what the implications are for CKD-MBD/ PO4 and cardiovascular disease and patient survival.
A number of sentences would be best shortened or divided into two sentences.
Author Response
Point-to-point reply to Reviewer 2 - Manuscript nutrients-2460417
The primary focus of this review as outlined in the abstract is that there are special issues required regarding appropriate control of PO4 in ESKD patients on peritoneal dialysis and that the are very few studies that address directly the issue of PO4 control in PD. This reviewer endorses these questions and would support the importance of dealing with these issues.
Recommend substantial shortening of the general discussion of pathophysiology of CKD-MBD, rather focus on the impact of an increased PO4 burden in CKD (and the physiologic response to this burden) and the evidence that supports the clinical consequences of this higher PO4 burden.
We have shortened the general discussion on the pathophysiology of CKD-MBD and have focused on the response to and the consequences of an increased phosphate burden, as suggested.
A detailed discussion of the evidence used to support PO4 binder therapy in general for PD patients and any specific evidence (or lack there of) for specific target serum PO4 levels.
We have added a discussion regarding the evidence supporting the use of phosphate binders in PD patients in the “Conclusions” section and changed the title of this section in “Discussion”. We have also underlined the lack of specific targets for phosphate serum levels in PD patients in paragraph 5.
Detail the areas for which evidence is lacking and why such evidence developed specifically in PD patients is important.
We have detailed the area for which evidence is lacking adding a discussion about the need for specific serum phosphate targets, because this could improve clinical management and then outcomes of PD patients.
Expand the discussion of the impact of PD prescription and of PD transport properties on PO4 clearance and the implications for treatment.
We have added new data and concepts on the topic as suggested by the Reviewer.
Expand the discussion of impact of diet restriction on protein malnutrition.
We have expanded the discussion of this point as suggested.
Include an acknowledgement that active vitamin D therapy has an impact on PO4 absorption and what the implications are for CKD-MBD/ PO4 and cardiovascular disease and patient survival.
We have added a discussion on the relationship between vitamin D deficiency and cardiovascular risk and on the role of active vitamin D treatment on phosphorous intestinal absorption.
Comments on the Quality of English Language: A number of sentences would be best shortened or divided into two sentences.
We have modified some sentences as suggested by the Reviewer.

Round 2
Reviewer 2 Report
Revised manuscript now serves as a comprehensive review of phosphorus metabolism and bone and mineral disease pathophysiology and management specifically as it relates to peritoneal dialysis. As such this treatment of the topic provides a very useful foundation for both investigators and practicing clinicians to address phosphorus in patients treated with peritoneal dialysis. It is particularly relevant that the authors highlight the gaps in our current state of knowledge. This acknowledgement may inform / stimulate future investigations.
Only one minor suggestions for the authors consideration. Might also acknowledge the potential role for systematic reviews and metaanalyses to summarize the field as new data emerges. I do think it might be appropriate to state (I defer to the authors) that the current body of evidence does not lend itself to meta-analysis due to the scarcity of PD specific trials. If the authors chose to add this brief topic, they might wish to add the number of RCT's in PD for each of the major conclusions that clinicians would want to best manage the PD population. If after a comprehensive systematic review, it turns out that there are just one or two RCT (or none at all) for major clinical questions, meta-analysis does not add anything.
Short of adding any discussion of the utility of meta-analysis, this reviewer would recommend the authors add a brief acknowledgment of their search strategy that enables a claim that the evidence presented was as comprehensive as the published literature allows and was unbiased and accurately reflect the science and not a selected view of the science. I would argue that this is a relatively important point and would be served by a description of the search strategies and the languages of the literature included.
Author Response
Point-to-point reply to Reviewer 2 (Round 2) - Manuscript nutrients-2460417
Revised manuscript now serves as a comprehensive review of phosphorus metabolism and bone and mineral disease pathophysiology and management specifically as it relates to peritoneal dialysis. As such this treatment of the topic provides a very useful foundation for both investigators and practicing clinicians to address phosphorus in patients treated with peritoneal dialysis. It is particularly relevant that the authors highlight the gaps in our current state of knowledge. This acknowledgement may inform / stimulate future investigations.
We thank the Reviewer for his/her comment.
Only one minor suggestions for the authors consideration. Might also acknowledge the potential role for systematic reviews and metaanalyses to summarize the field as new data emerges. I do think it might be appropriate to state (I defer to the authors) that the current body of evidence does not lend itself to meta-analysis due to the scarcity of PD specific trials. If the authors chose to add this brief topic, they might wish to add the number of RCT's in PD for each of the major conclusions that clinicians would want to best manage the PD population. If after a comprehensive systematic review, it turns out that there are just one or two RCT (or none at all) for major clinical questions, meta-analysis does not add anything.
Short of adding any discussion of the utility of meta-analysis, this reviewer would recommend the authors add a brief acknowledgment of their search strategy that enables a claim that the evidence presented was as comprehensive as the published literature allows and was unbiased and accurately reflect the science and not a selected view of the science. I would argue that this is a relatively important point and would be served by a description of the search strategies and the languages of the literature included.
We thank the Reviewer for the suggestions. We have added information about the search strategy and included considerations on the results provided by the few studies published on the topic and on the potential utility of systematic reviews and meta-analyses in the field as new data emerges.
